# Effect of Dune Sand on Drying Shrinkage Cracking of Fly Ash Concrete

**Euibae Lee [1], Jeongwon Ko [1], Jaekang Yoo [1], Sangjun Park [2] and Jeongsoo Nam [3,\*]**

1    Daewoo Institute of Construction Technology, DAEWOO E&C, Suwon 16297, Korea;
     mir2468@naver.com (E.L.); Jeongwon.ko@daewooenc.com (J.K.); jaekangyoo@gmail.com (J.Y.)
2    Department of Building Construction, Korea Polytechnics, Gumi 39257, Korea; concrete@kopo.ac.kr
3    Department of Architectural Engineering, Chungnam National University, Daejeon 34134, Korea
\*    Correspondence: j.nam@cnu.ac.kr; Tel.: +82-42-821-5629

**Abstract:** In this study, the drying shrinkage cracking characteristics of concrete containing 30% fly ash were investigated for various dune sand (DS) replacements and unit water contents. In the results of compressive strength, the mixture with a DS-to-fine aggregate (DS/FA) ratio of 10% showed the highest value, which was 17.6% higher than the lowest value. However, in the results of restrained drying shrinkage cracking, the mixtures with a DS/FA ratio of 20% showed the highest crack resistance, which was 24% higher than the lowest crack resistance. Therefore, it is necessary to consider DS replacement according to the required performance of concrete. The restraint effect factors of the aggregates were analyzed based on the relationship between the volume change of each aggregate and the change in the net time to cracking. The restraint effect factor of crushed sand was 0.71–0.98 and that of DS was 0.56–0.90 when that of the coarse aggregate (CA) was 1.

**Keywords:** dune sand; fly ash; drying shrinkage cracking; net time to cracking; restraint effect factor

## 1. Introduction

It is essential for concrete structures to continue to perform their intended functions during their expected service lives. It follows that concrete must be able to withstand the processes of deterioration to which it can be expected to be exposed [1]. That is, it is necessary to make concrete durable, and an appropriate concrete mix for the specific environment to which the concrete structure is exposed should be determined [2–4].

Many countries on the Arabian Peninsula have continuously invested in the development of infrastructure. In particular, they have invested the most significantly in the construction of petrochemical and power plants, and global contractors have been conducting many related projects. These structures are mainly located along the coast of the Arabian Peninsula, and the groundwater and subsoils in coastal areas are likely to be contaminated with various salts that are aggressive to concrete [5]. Therefore, corrosion induced by both chlorides from sea water and chemical attacks should be considered when designing concrete for durable coastal structures.

Various international and regional codes provide guidelines for durable concrete mix design. Among them are the BS (EN) codes and the ACI codes, which are widely used internationally. On the Arabian Peninsula, specifications based on the BS codes are applied in most projects. According to the BS codes, the maximum water/cement ratio, minimum cement content, and minimum compressive strength are limited for the design of durable concrete structures in special environments. In particular, to address the chemical attack of underground structures, the type of cement is limited to prevent the concrete from deteriorating. Instead of ordinary Portland cement (OPC), it is recommended to use sulfate-resisting Portland cement or a combination with a large amount of fly ash or ground-granulated blast-furnace slag.

In addition, the climatic and material conditions should be considered in order to ensure the consistent quality of the structural concrete on the Arabian Peninsula. First, the Arabian Peninsula lies in hot and dry desert climate zones. This area has a higher risk of concrete cracking caused by increased plastic and drying shrinkage. As a raw material for concrete, dune sand (DS) is used as fine aggregate (FA) for concrete in this area. DS has a spherical grain shape and a smooth surface, and it can help improve the grading of crushed sand (CS). However, most of the particles in DS are smaller than 0.6 mm. Thus, the excessive use of DS may reduce workability by increasing viscosity [6]. Many studies have been conducted on the characteristics of concrete that used DS as an aggregate, and most of them have been focused on strength and workability [7–12]. However, as previously mentioned, a high risk of cracking is evident on the Arabian Peninsula due to drying shrinkage. Therefore, it is necessary to examine the characteristics of drying shrinkage cracking of concrete in this area.

As previously mentioned, sulfate-resisting Portland cement or a combination with fly ash or ground-granulated blast-furnace slag should be used as a binder for structures in coastal areas on the Arabian Peninsula. In order to review the drying shrinkage cracking characteristics of concrete for structures in this area, an experimental study considering binders other than OPC is required. Some researchers have studied the characteristics of geopolymers made with fly ash or ground-granulated blast-furnace slag as a binder and DS [13,14]. The properties of blended cement mixtures incorporating ground DS and ground-granulated blast-furnace slag have also been studied [15,16]. However, the material conditions and shrinkage cracking properties of concrete for coastal structures on the Arabian Peninsula were not considered in these investigations. Hence, the objective of the present research was to investigate the drying shrinkage cracking of concrete containing a binder with the consideration of durability mix design conditions for costal structures on the Arabian Peninsula and DS. For concrete mixtures, a combination with OPC and fly ash was used as a binder, and various volumetric DS-to-FA (DS/FA) ratios and unit water contents were considered. In addition, the effect of each aggregate on the drying shrinkage cracking characteristics was quantitatively analyzed.

## 2. Experimental Plan and Method

### 2.1. Experimental Plan

Concrete properties and limiting values to resist corrosion induced by chlorides from sea water and chemical attacks are recommended in BS 8500-1 [17]. The minimum compressive strength, maximum water-to-binder (W/B) ratio, and minimum cement content are also suggested for various types of cement and combinations. In this study, a combination of OPC and fly ash was used as a binder for concrete. In order to satisfy the condition of sulfate-resisting cement, the fly ash replacement is required to be in the range of 25–35% according to BS 8500-1. Therefore, the replacement ratio of fly ash was set to 30% in this study. In addition, BS 8500-1 recommends W/B ratios from 0.35 to 0.55 in increments of 0.05. In this study, the water/cement ratio of 0.4 was set. The target slump of concrete was $180 \pm 25$ mm. Regarding the mixture variables, the unit water content was set to 170 and 160 kg/m$^3$, and the DS/FA ratio was set to 10%, 20%, 40%, and 60%.

### 2.2. Materials and Mixture Proportions

The cement used was ordinary Portland cement that had a density of 3.15 g/cm$^3$ and fineness of 3440 cm$^2$/g. The density and fineness of the fly ash used were 2.31 g/cm$^3$ and 3135 cm$^2$/g, respectively. The chemical analysis data of the cement and fly ash are shown in Table 1.

As the fine aggregate, a binary sand with replacements of DS was used, and the morphologies of CS and DS are shown in Figures 1 and 2, respectively. CS is composed of relatively large particles with angular shapes. However, DS is composed of relatively small particles with spherical shapes. Table 2 shows the physical properties of DS, CS, and

coarse aggregate (CA). The maximum size of CA was 20 mm. Table 3 shows the chemical composition of DS, and it mainly consists of $SiO_2$ and CaO.

**Table 1.** Chemical composition of cement and fly ash.

| Material | Chemical Composition (%) | | | | | | | |
|---|---|---|---|---|---|---|---|---|
| | $SiO_2$ | $Al_2O_3$ | $Fe_2O_3$ | CaO | MgO | $SO_3$ | $K_2O$ | LOI |
| Cement | 21.49 | 5.66 | 3.25 | 61.45 | 2.29 | 2.38 | 1.00 | 2.33 |
| Fly ash | 57.10 | 18.76 | 6.96 | 10.27 | 1.56 | 0.61 | 0.89 | 3.84 |

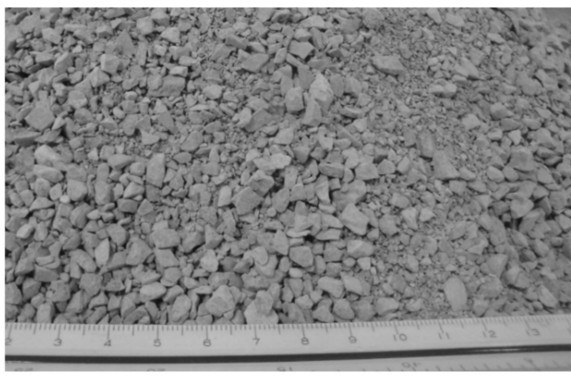

**Figure 1.** CS appearance.

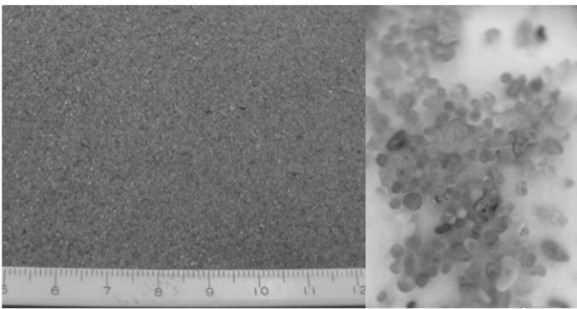

**Figure 2.** DS appearance.

**Table 2.** Physical properties of aggregates.

| Aggregate | Physical Properties | | |
|---|---|---|---|
| | Density (g/cm$^3$) | Absorption Ratio (%) | Fineness Modulus |
| DS | 2.61 | 1.19 | 0.7 |
| CS | 2.61 | 1.53 | 3.6 |
| CA | 2.70 | 0.77 | - |

**Table 3.** Chemical composition analysis results of DS.

| Chemicals | $SiO_2$ | CaO | $Al_2O_3$ | MgO | $Fe_2O_3$ | $K_2O$ |
|---|---|---|---|---|---|---|
| % | 47.1 | 38.8 | 5.65 | 3.03 | 2.68 | 1.44 |

Figure 3 shows the particle size distribution of the binary sand according to the DS/FA ratios. Neither CS nor DS was within the standard particle size range recommended by KS F 2527 [18] or ASTM C 33 [19].

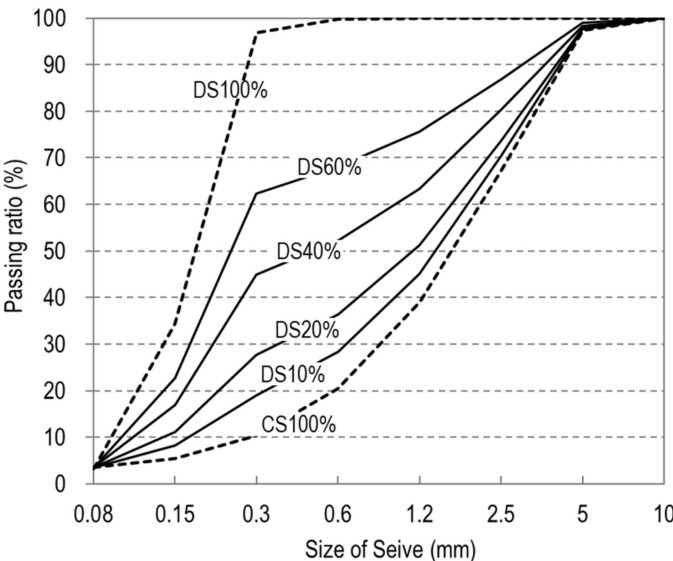

**Figure 3.** Particle size distribution of sand with DS replacement.

The mixture proportions of concrete used in this study are shown in Table 4. The sand-to-aggregate (S/a) ratio and chemical admixture (AD) content were adjusted through trial mixing test. In the case of a mixture using 100% CS, it was not possible to achieve the target slump despite the increase in AD content. The mixtures with 10% or higher DS/FA ratios ensured that the target slump was reached in this study.

**Table 4.** Concrete mixture proportions.

| Mixture Identity | | Slump (mm) | W/B | S/a [3] | Unit Weight (kg) | | | | | | AD [10] (bw%) |
|---|---|---|---|---|---|---|---|---|---|---|---|
| | | | | | W [4] | C [5] | F [6] | CS [7] | DS [8] | CA [9] | |
| W170 [1] | D10 [2] | | | 0.60 | 170 | 298 | 128 | 929 | 103 | 712 | 0.70 |
| | D20 | | | 0.50 | 170 | 298 | 128 | 688 | 172 | 890 | 0.65 |
| | D40 | | | 0.45 | 170 | 298 | 128 | 464 | 310 | 978 | 0.70 |
| | D60 | 180 ± 25 | 0.4 | 0.42 | 170 | 298 | 128 | 289 | 433 | 1032 | 0.80 |
| W160 | D10 | | | 0.65 | 160 | 280 | 120 | 1035 | 115 | 641 | 0.90 |
| | D20 | | | 0.55 | 160 | 280 | 120 | 778 | 195 | 824 | 0.80 |
| | D40 | | | 0.47 | 160 | 280 | 120 | 499 | 333 | 970 | 0.90 |
| | D60 | | | 0.44 | 160 | 280 | 120 | 311 | 467 | 1025 | 1.00 |

[1] W160: water content 160 kg/m³, [2] D10: DS/FA ratio 10%, [3] S/a: sand-to-aggregate ratio, [4] W: water, [5] C: ordinary Portland cement, [6] F: fly ash, [7] CS: crushed sand, [8] DS: dune sand, [9] CA: coarse aggregate, [10] AD: chemical admixture.

### 2.3. Test Method

Compressive strength and split tensile strength were tested with three specimens at ages of 3, 7, 14, and 28 days according to KS F 2405 [20] (ASTM C 39 [21]) and KS F 2423 [22] (ASTM C 496 [23]).

The ring-type restrained shrinkage cracking test proposed by ASTM C 1581 [24] was conducted to evaluate the drying shrinkage cracking characteristics of concrete. To fabricate specimens, the mixed concrete was sieved to remove CA that exceeded 13 mm. The concrete was poured into a test mold in two layers, and each layer was compacted with tamping rods. Each fabricated specimen was moved into a constant temperature and humidity chamber at a temperature of 23 ± 2 °C and a relative humidity of 50 ± 4%. Regarding the inner ring of the specimen, the bolt was removed so that the ring could not be fixed to the bottom plate, and the strain gauge attached to the ring was connected to a data logger. The specimen was covered with polyethylene film to prevent drying. After one day, the polyethylene film and outer ring mold were removed, and the drying was initiated. Two specimens of restrained drying shrinkage cracking test were made for each mixture.

ASTM C 1581 proposes the net time to cracking and stress rate as measures to evaluate the cracking risk of concrete mixture. The net time to cracking represents the time difference between the onset of specimen drying and the time of cracking. The time of cracking refers to a point at which a strain that has slowly increased suddenly drops. The stress rate was calculated using Equations (1) [24] and (2) [24]. In this instance, the strain rate factor in Equation (1) was calculated based on the regression analysis of the measured data. In this study, regression analysis was conducted using the nonlinear regression analysis program (NLREG) software.

$$\varepsilon_{net} = \alpha\sqrt{t} + k \tag{1}$$

where $\varepsilon_{net}$ is the net strain (m/m), $\alpha$ is the strain rate factor for each strain gauge $((\text{m/m})/\text{day}^{1/2})$, $t$ is the elapsed time (days), and $k$ is the regression constant.

$$q = \frac{G|\alpha_{avg}|}{2\sqrt{t_r}} \tag{2}$$

where $q$ is the stress rate in each test specimen (MPa/day), $G$ is 72.2 GPa, $|\alpha_{avg}|$ is the absolute value of the average strain rate factor $((\text{m/m})/\text{day}^{1/2})$, and $t_r$ is the elapsed time at cracking (days).

Additionally, drying shrinkage stress was calculated using Equation (3) [25] in this study.

$$\sigma_c(t) = \varepsilon_s E_s \frac{R_{IS}^2}{R_{OS}^2} \frac{\left(R_{OS}^2 - R_{IS}^2\right)}{(1 + v_s)R_{OS}^2 + (1 - v_s)R_{IS}^2} \frac{\left(R_{OC}^2 + R_{IC}^2\right)}{\left(R_{OC}^2 - R_{IC}^2\right)} \tag{3}$$

where $\sigma$ is the shrinkage stress of concrete (MPa), $\varepsilon_s$ is the strain of the steel ring ($\times 10^{-6}$), *Es* is the elastic modulus of the steel ring (MPa), $v_s$ is the Poisson's ratio of the steel ring, $R_{IS}$ is the inner diameter of the steel ring (mm), $R_{OS}$ is the outer diameter of the steel ring (mm), $R_{IC}$ is the inner diameter of concrete (mm), and $R_{OC}$ is the outer diameter of concrete (mm).

## 3. Result and Discussion

### 3.1. Compressive Strength and Tensile Strength

Figure 4 shows the histories of compressive strength development. In the results of compressive strength at 28 days, the compressive strengths of the W170 mixtures were in the range of 39.9–45.7 MPa. The D40 mixture showed the lowest compressive strength. The D10 mixture showed the highest compressive strength, which was 14.4% higher than that of the DS40 mixture. The W160 mixtures also showed similar results to those of the W170 mixtures. The D10 mixture showed the highest compressive strength of 47.0 MPa. The lowest compressive strength was 39.9 MPa, as shown in the results of the D60 mixture. The compressive strength of the D10 mixture was 17.6% higher than that of the D60 mixture. These results indicate that more than 10% difference in compressive strength could occur depending on the DS replacement ratio.

Some researchers [7,8,26] reported that the compressive strength of concrete using dune sand reduced as the DS replacement ratio increased. This could be attributed to the decrease in bond strength between the cement paste and DS due to the smooth surfaces and round shape of the DS particles [27]. In contrast, other researchers [9–11] reported that the strength increased until the DS replacement ratio attained a certain value but that it decreased at higher replacement ratios. Luo et al. [8] proposed that the increase in strength could be due to high aggregate compactness. Based on the results of previous studies, DS could have an effect on both the compactness of aggregates and bond strength between DS and the cement paste. Moreover, the effect of DS on both sides has opposite effects on compressive strength.

In the mixture conditions of this study, the DS/FA ratio of 10% was optimal for compressive strength. Furthermore, when the DS/FA ratio was more than 10%, the decrease in bond strength between DS and the cement paste could have a greater effect than that of the compactness of the aggregate.

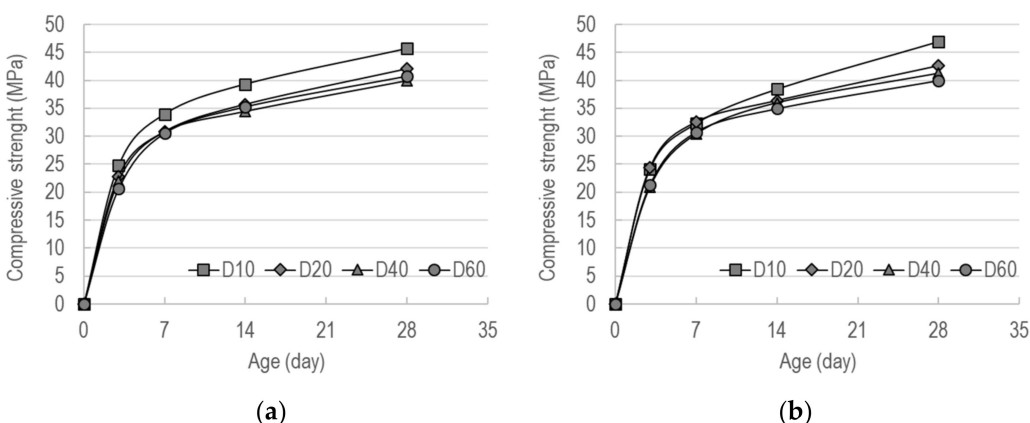

(a)        (b)

**Figure 4.** Compressive strength histories: (**a**) W170 mixtures and (**b**) W160 mixtures.

Figure 5 shows the histories of split tensile strength development. Similar to the compressive strength, the D10 mixture showed the highest tensile strength regardless of unit water content. The highest tensile strengths were 30.1% and 21.6% higher than the lowest in the W170 mixtures and W160 mixtures, respectively.

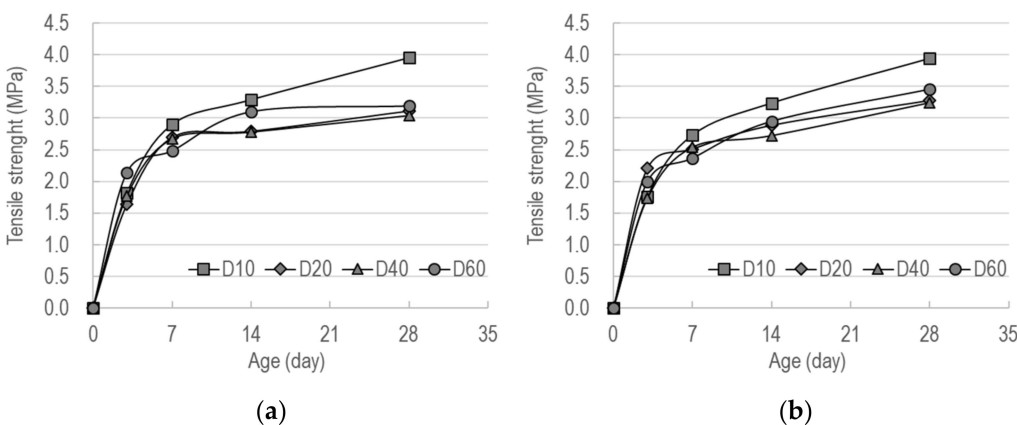

(a)        (b)

**Figure 5.** Split tensile strength histories: (**a**) W170 mixtures and (**b**) W160 mixtures.

### 3.2. Net Time to Cracking

Figures 6 and 7 show the history of the inner ring strain measured after the onset of drying. In general, as shown in Figure 6a, the strain gradually increases and then suddenly drops greatly, and the point of change is regarded as the cracking point. However, in this study, other types of strain history were also observed, as shown in Figure 6b. Each specimen was observed periodically every day to confirm the exact time of cracking.

The cracks occurring on the sides of each specimen could be classified into three types, as shown in Figure 8: a crack across the entire height of the specimen, multiple cracks across the height of the specimen, and a short crack in the upper or lower part of the specimen. These cracks occurred in one or two places on the specimens. Two strain gauges were symmetrically attached to the inner ring, and the locations at which the cracks occurred were either near the strain gauge or between the strain gauges. The pattern, number, and locations of the cracks could have affected the strain gauge; therefore, the strain history curve could appear in various ways. In this study, the net time to cracking was determined

by considering the time at which cracking was confirmed by regular visual observation and the time at which abnormal strain deformation occurred.

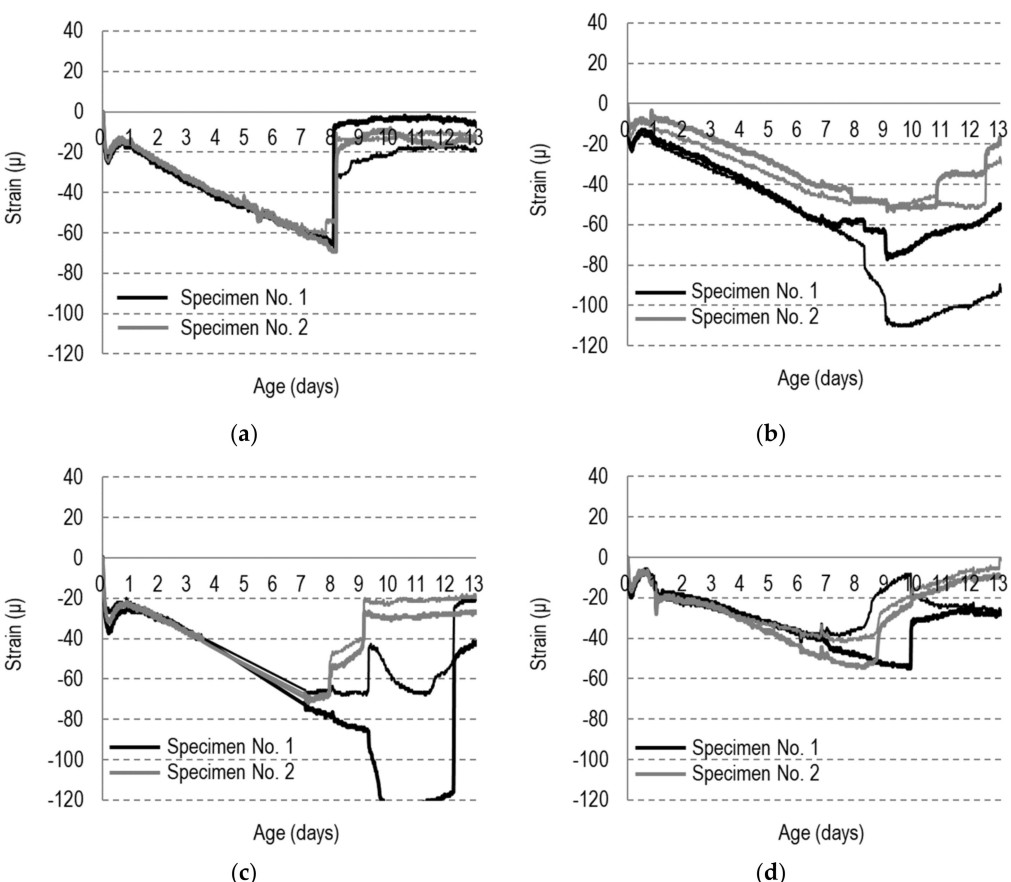

**Figure 6.** Strain history of inner ring (W170 mixtures): (**a**) D10, (**b**) D20, (**c**) D40, and (**d**) D60.

Figure 9 shows the net time-to-cracking results. The net times to cracking of the W170 mixtures range from 8.10 to 8.67 days, and the D20 mixture shows the maximum net time to cracking. Conversely, the D10 mixture yields the minimum net time to cracking, which is approximately 93% of that of the value for the D40 mixture. The net times to cracking of the W160 mixtures range from 8.13 to 10.04 days. The D20 mixture shows the maximum net time to cracking, and the D40 mixture exhibits the minimum net time to cracking, which is approximately 81% of the maximum net time to cracking.

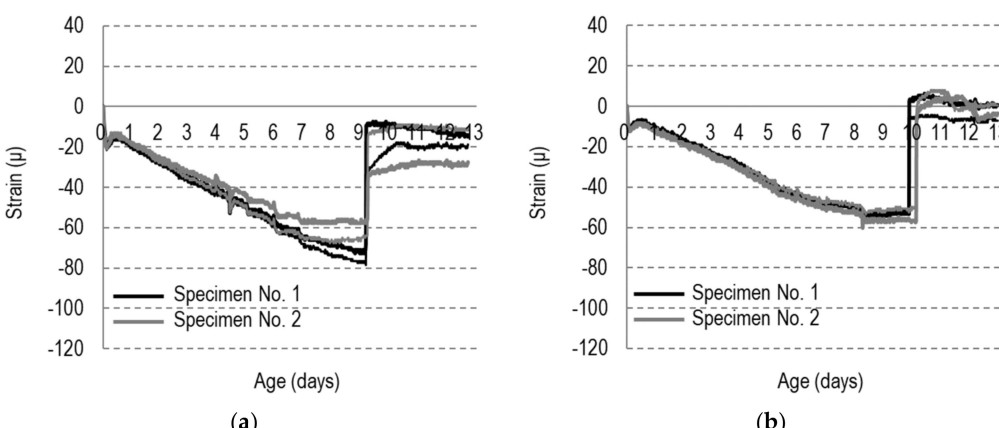

**Figure 7.** *Cont.*

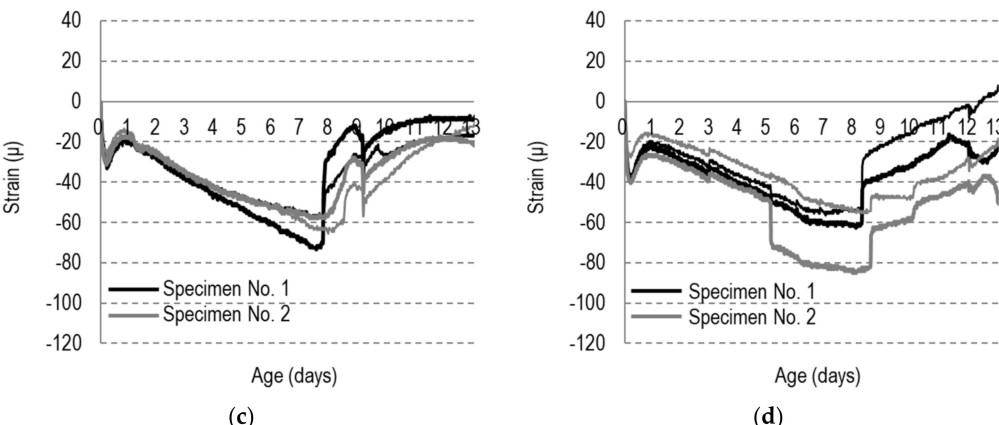

(c)                    (d)

**Figure 7.** Strain history of inner ring (W160 mixtures): (**a**) D10, (**b**) D20, (**c**) D40, and (**d**) D60.

The average net time to cracking is 8.45 days for the W170 mixtures and 8.95 days for the W160 mixtures, indicating that the W160 mixtures have the highest crack resistance. In general, it is known that a concrete mixture that has a low water content has lower risks of drying shrinkage and cracking.

The D10, D20, D40, and D60 mixtures yielded average net times to cracking of 8.65, 9.35, 8.35, and 8.45 days, respectively, indicating that the D20 mixture had the highest crack resistance.

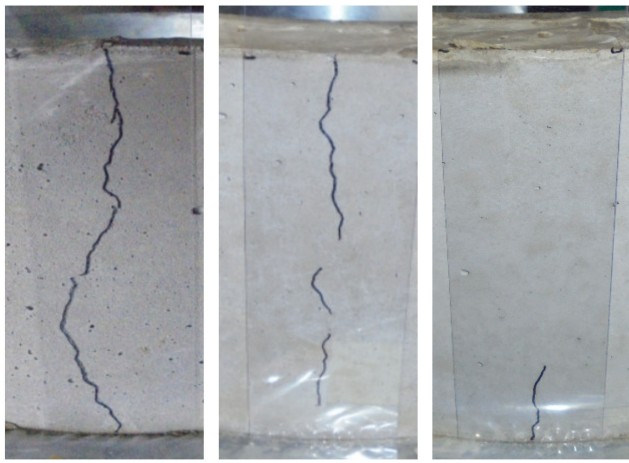

**Figure 8.** Examples of shrinkage cracking pattern.

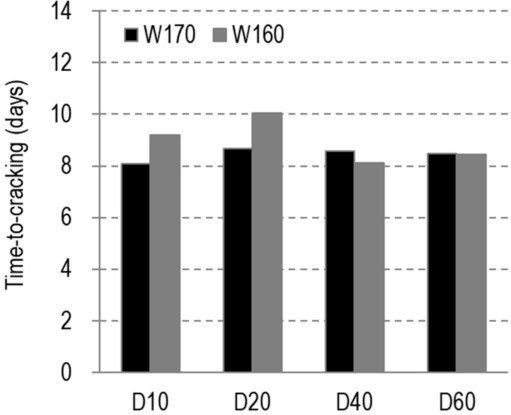

**Figure 9.** Net time to cracking.

### 3.3. Shrinkage Strain of Inner Ring and Shrinkage Stress at Cracking

Figure 10 shows the shrinkage strain of the inner ring immediately before cracking for each mixture.

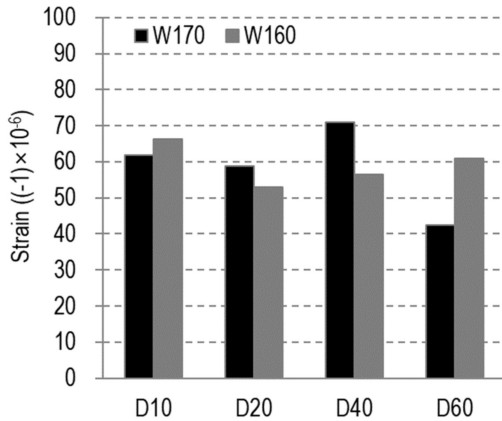

**Figure 10.** Drying shrinkage strain at cracking.

The shrinkage strains of the W170 mixtures range from 43 to $71 \times 10^{-6}$, and the D40 mixture shows the maximum shrinkage strain. The D60 mixture exhibits the minimum shrinkage strain, which is approximately 60% of the shrinkage strain of the D40 mixture. The shrinkage strains of the W160 mixtures range from 53 to $66 \times 10^{-6}$, and the D10 mixture exhibits the maximum shrinkage strain. The D20 mixture shows the minimum shrinkage strain, which is approximately 80% of the maximum shrinkage strain.

Figure 11 presents the shrinkage stress of each mixture before cracking was calculated using Equation (3). The shrinkage stress results show the same tendency as the shrinkage strain results. The shrinkage stresses of the W170 and W160 mixtures range from 3.3 to 5.6 MPa and 4.2 to 5.2 MPa, respectively. The shrinkage stress and net time to cracking were compared, and it was confirmed that there was no close relationship between them.

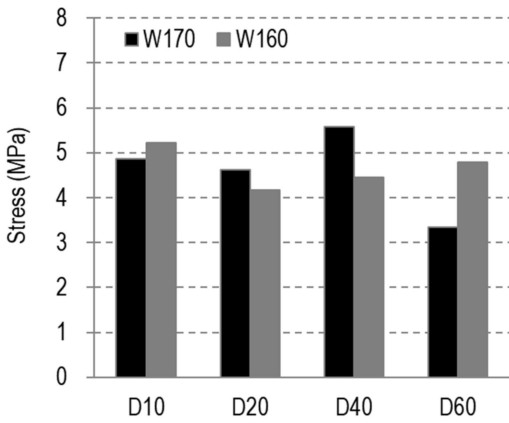

**Figure 11.** Drying shrinkage stress at cracking.

Figure 12 compares the splitting tensile strength development history with the shrinkage stress at cracking. Based on the measured data, the splitting tensile strength development history was derived using Equation (4), which is presented in ACI 209 [28].

$$f_{ts}(t) = \left[ \frac{t}{a + bt} \right] f_{ts}28 \qquad (4)$$

where $f_{ts}$ is the tensile strength at age t (MPa), *t* is the age (days), $f_{ts}28$ is the tensile strength at age 28 days (MPa), and *a* and *b* are constants.

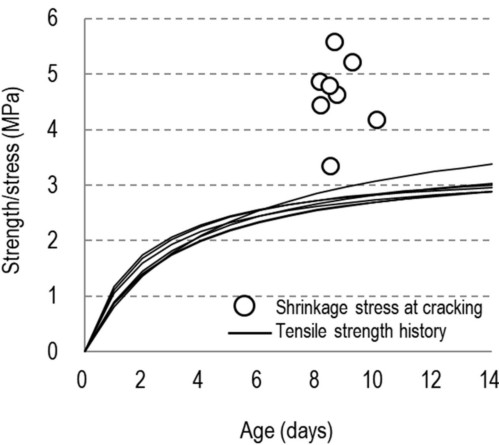

**Figure 12.** Development of tensile strength and shrinkage stress at cracking.

The comparison results show that the shrinkage stress at cracking is higher than the splitting tensile strength of concrete developed at the same time. Thus, predicting the cracking of concrete simply based on the splitting tensile strength and shrinkage stress of concrete may yield inaccurate results. This inaccuracy may occur because the cracking of restrained concrete is not caused by shrinkage stress alone but, rather, is attributable to the complex actions of various factors, such as the shrinkage rate, tensile creep, and tensile strength [29].

*3.4. Stress Rate*

Figure 13 presents the stress rate calculation results for each mixture. The stress rates of the W170 mixtures range from 0.21 to 0.32 MPa/day, and the D10 mixture shows the maximum stress rate. Conversely, the D60 mixture exhibits the minimum stress rate, which is approximately 66% of the value for the D40 mixture. The stress rates of the W160 mixtures range from 0.25 to 0.31 MPa/day. The minimum stress rate is observable for the D20 mixture, while the other mixtures (except the D20 mixture) show the same stress rates.

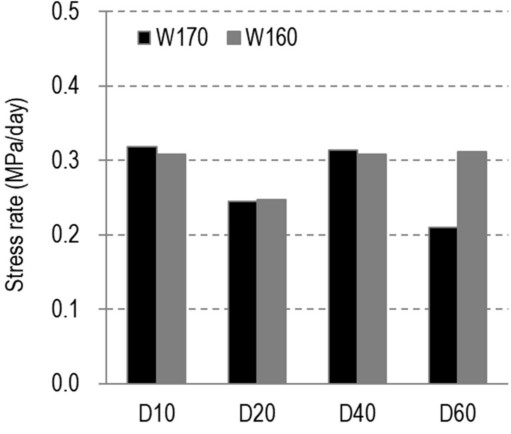

**Figure 13.** Stress rate.

The average stress rate is 0.27 MPa/day for the W170 mixtures and 0.29 MPa/day for the W160 mixtures, indicating that the W170 mixtures had the highest crack resistance. The D10, D20, D40, and D60 mixtures yielded average stress rates of 0.31, 0.25, 0.31, and 0.26 MPa/day, respectively. Correspondingly, the D20 mixture produced the highest crack resistance.

Figure 14 compares the net time to cracking with the stress rate. As the net time to cracking increases, the stress rate tends to decrease. However, the correlation between the net time to cracking and stress rate is not high. As previously mentioned, because the

cracking of restrained concrete is attributable to the complex actions of various factors, the correlation between the net time to cracking, which is the final result of cracking, and the stress rate, which is one of the causes of cracking, could be low.

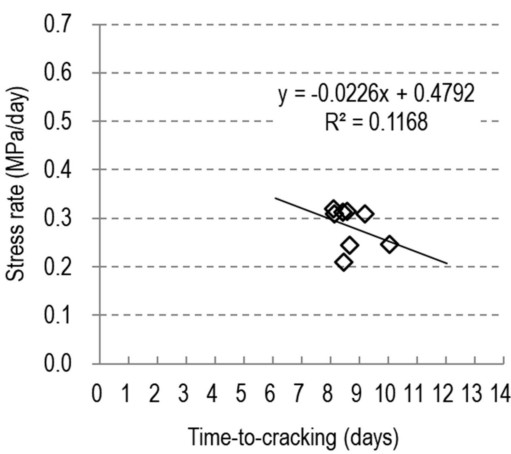

**Figure 14.** Relationship between net time to cracking and stress rate.

### 3.5. Effect of Aggregates on Drying Shrinkage Cracking

Drying shrinkage cracking typically decreases as the volume of cement paste decreases, while that of aggregate increases [30–33]. However, mixtures with the same W/B ratios and unit water contents have the same cement paste volume, while the aggregate volume ratio (weight ratio) varies depending on the aggregate types used and the sand-to-aggregate ratio. Regarding the mixture proportions used in this study, when the DS/FA ratio increased with the same unit water content, CS decreased and CA increased. In addition, as the aggregate volume fraction changed, the net time to cracking and stress rate also changed. That is, the change in the volume fraction of the aggregate also affected the shrinkage cracking characteristics of the concrete.

As such, the effect of the change in the volume of each aggregate type on the net time to cracking, which is a measure of shrinkage cracking, was also examined in this study. First, based on the mixture with the highest net time to cracking among the mixtures with the same unit water content, the volume change of each aggregate and the change in net time to cracking were calculated for the other mixtures. The volume change of each aggregate and the change in net time to cracking were calculated based on the D20 mixture for the W170 and W160 mixtures. The results are shown in Table 5.

**Table 5.** Aggregate volume change and the change in net time to cracking.

| Mixture Identity | | Change in Aggregate Volume ($\ell$) | | | Change in Net Time to Cracking |
|---|---|---|---|---|---|
| | | CS | DS | CA | |
| W170 | D10 | 92 | −26 | −66 | −0.57 |
| | D40 | −86 | 53 | 33 | −0.08 |
| | D60 | −153 | 100 | 53 | −0.20 |
| W160 | D10 | 98 | −31 | −68 | −0.85 |
| | D40 | −107 | 53 | 54 | −1.92 |
| | D60 | −179 | 104 | 75 | −1.61 |

In addition, the relationship between the change in each aggregate's volume and the change in net time to cracking is proposed in Equation (5) [34].

$$\Delta T_{cr} = a \cdot \Delta DS + b \cdot \Delta CS + c \cdot \Delta CA \tag{5}$$

where $\Delta T_{cr}$ is the change in the net time to cracking (days); $\Delta DS$ is the DS volume change ($\ell$); $\Delta CS$ is the CS volume change ($\ell$); $\Delta CA$ is the CA volume change ($\ell$); *a* is the restraint

effect factor of DS affecting the net time to cracking (days·ℓ); *b* is the restraint effect factor of CS, which affects the net time to cracking (days·ℓ); and *c* is the restraint effect factor of CA, which affects the net time to cracking (days·ℓ).

In general, DS has a smooth surface and a round shape. Conversely, CS and CA have rough surfaces and angular shapes. The physical properties of these aggregates may affect the adhesion and binding force between the cement paste and aggregates. That is, an increase in the volume of DS will reduce the restraint effect of the aggregate, and increases in the volumes of CS and CA will increase the restraint effect. In addition, as the size of the aggregate increases, the restraint effect increases, reducing the drying shrinkage [32]. Therefore, the restraint effect of each aggregate can be assumed to adhere to the order of $0 < a < b < c$.

Finally, after setting the largest factor c to 1, subject to the restraint effect factor condition of each aggregate type, the restraint factor a was calculated based on the substitution of the second largest factor b (0.99–0.01) in Equation (5). Among the calculated factor values, the ranges of the factors that met the assumption presented above were calculated. Figures 15 and 16 show the results.

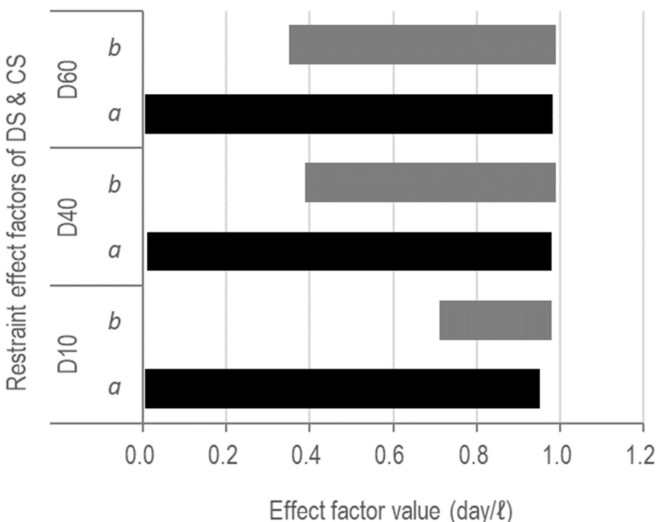

**Figure 15.** Analysis results of restraint effect factor of DS and CS (W170 mixtures); *a* is the restraint effect factor of DS and *b* is the restraint effect factor of CS.

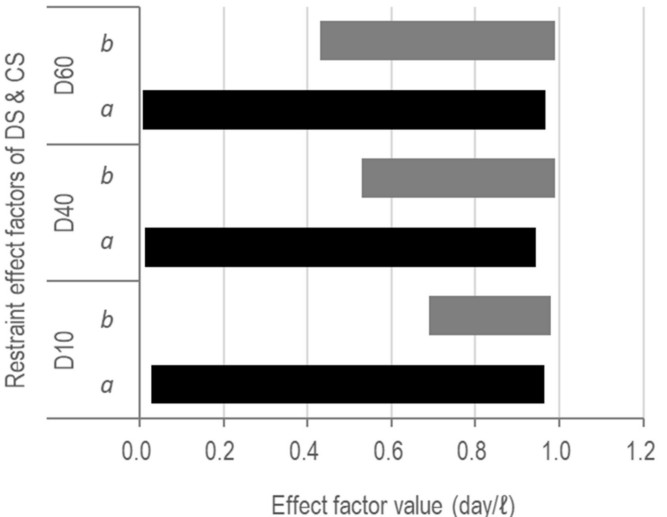

**Figure 16.** Analysis results of restraint effect factor of DS and CS (W160 mixtures); *a* is the restraint effect factor of DS and *b* is the restraint effect factor of CS.

In the case of the W170 mixtures, when the restraint effect factor of CA was 1, the restraint effect factor of CS ranged from 0.71 to 0.98 days/$\ell$ and that of DS ranged from 0.56 to 0.95 days/$\ell$. In the case of the W160 mixtures, the restraint effect factor of CS ranged from 0.69 to 0.98 days/$\ell$ and that of DS ranged from 0.45 to 0.92 days/$\ell$. In all the mixtures, the restraint effect factor of CS ranged from 0.71 to 0.98 days/$\ell$ and that of DS ranged from 0.56 to 0.90 days/$\ell$.

## 4. Conclusions

This study experimentally evaluated the drying shrinkage cracking characteristics of FA concrete for various unit water contents and DS/FA ratios. The results can be summarized as follows:

(1) In the mixture conditions of this study, the DS/FA ratio of 10% was optimal for compressive strength and split tensile strength. Depending on the DS/FA ratio, the difference in strength could occur by more than 10%.

(2) In the average net time-to-cracking results, the mixtures with a DS/FA ratio of 20% showed the highest crack resistance. The risk of drying shrinkage cracking could be improved by up to about 24% by optimal replacement of DS.

(3) The correlations of the shrinkage strain, shrinkage stress, and strain rate with the net time to cracking were not close in this study. In addition, the shrinkage stress at cracking was higher than the splitting tensile strength of the concrete. Therefore, to predict the shrinkage cracking of concrete, the complex actions of various factors must be considered.

(4) The optimum DS/FA ratio for the strength and crack resistance of concrete is different. Therefore, it is necessary to consider DS replacement according to the required performance of concrete.

(5) The restraint effect factors of the aggregates were analyzed based on the relationship between the volume change of each aggregate type and the change in net time to cracking. The restraint effect factor of CS was found to range from 0.71 to 0.98 and that of DS from 0.56 to 0.90 when the restraint effect factor of CA was 1 for all the mixtures.

**Author Contributions:** Conceptualization, E.L.; methodology, J.N.; validation, J.K., J.Y. and S.P.; formal analysis, E.L. and J.N.; investigation, E.L., J.K., J.Y. and S.P.; data curation, E.L.; writing—original draft preparation, E.L.; writing—review and editing, J.N.; supervision, S.P.; project administration, E.L. All authors have read and agreed to the published version of the manuscript.

**Funding:** This research received no external funding.

**Institutional Review Board Statement:** Not applicable.

**Informed Consent Statement:** Not applicable.

**Data Availability Statement:** Not applicable.

**Conflicts of Interest:** The authors declare no conflict of interest.

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
