# Peer review of "Effect of Dune Sand on Drying Shrinkage Cracking of Fly Ash Concrete"

_applsci, doi:10.3390/app12063128_

Round 1

Reviewer 1 Report

  • Some redundant statement should be removed” Featured Application: Authors are encouraged to provide a concise description of the specific application or a potential application of the work. This section is not mandatory”
  • The article still needs several grammatical and syntax improvements. Use of English service center is recommended.
  • The abstract should be written more clearly. Majority of the qualitative statements should be modified for quantified result comparisons.
  • The introduction needs to be revised for higher quality language. The authors mentioned some works without stating about the contributions, pros and cons and the how the current work would address. The language should be improved.
  • The authors mentioned “it is necessary to make the concrete durable, and an appropriate concrete mix for the specific environment in which the concrete structure is exposed should be determined”. The following references should be added for this statement.
    • Experimental investigation of sound transmission loss in concrete containing recycled rubber crumbs.
    • Compressive behavior of concrete under environmental effects. IntechOpen.
    • Temperature and humidity effects on behavior of grouts. Advances in concrete construction, 5(6), 659.
  • The workability should be defined, and how the Dune sand affect the workability?
  • The equations previously developed should be clearly referenced.
  • All the figures including figure 15, and figure 16 should have clear legends, descriptive axis labels and appropriate explanation.
  • How some of the specimens restrain effect factor are started form a specified value.
  • The reason for monitoring the time to crack should be justified and explained
  • The restrain effect factor should be clearly explained.

Author Response

Thank you for your comments for the paper.

We considered carefully your comments and our answers are the follows.

  1. The article still needs several grammatical and syntax improvements. Use of English service center is recommended.

: As you commented, English grammar or syntax may not be satisfactory, but, in order to write this article, we received the service of 00000, a professional English translation center. Please take this into consideration.

  1. The abstract should be written more clearly. Majority of the qualitative statements should be modified for quantified result comparisons.

: As you commented, the abstract was revised to be more quantitative.

  1. The introduction needs to be revised for higher quality language. The authors mentioned some works without stating about the contributions, pros and cons and the how the current work would address. The language should be improved. The authors mentioned “it is necessary to make the concrete durable, and an appropriate concrete mix for the specific environment in which the concrete structure is exposed should be determined”. The following references should be added for this statement.

“Experimental investigation of sound transmission loss in concrete containing recycled rubber crumbs.”

“Compressive behavior of concrete under environmental effects”. IntechOpen.

“Temperature and humidity effects on behavior of grouts”. Advances in concrete construction, 5(6), 659.

: The references you recommended were added.

  1. The equations previously developed should be clearly referenced.

: The papers introducing the equation are listed in References.

  1. All the figures including figure 15, and figure 16 should have clear legends, descriptive axis labels and appropriate explanation.

: As you commented, figures 15 and 16 were modified to show more clearly.

  1. How some of the specimens restrain effect factor are started form a specified value.

: In previous study, the equation of the relation between the change of aggregate volume and the change of net time to cracking was suggested. The restraint effect factor was used to quantitatively solve the relationship. As you know, drying shrinkage of concrete decreases as the restraint of aggregate increases. And we can understand the magnitude of restraint with aggregate type based on the conditions of size, shape and surface texture; 0 < a < b < c.

The change of aggregate volume can be found in mixture proportions. The change of net time to cracking that is a representative value of crack resistance easily calculated from the measurement data. But it is difficult to quantify the restraint effect of aggregate. So, in this study, the largest value that is the restraint effect factor of CA was assumed to be 1. And the value of restraint effect factor of DS was obtained by substituting the value of CS in sequence within a certain range. Finally, the values of restraint factor value of DS and CS could be obtained quantitatively, and the values are relative values to the CA values.

  1. The reason for monitoring the time to crack should be justified and explained

: ASTM C 1581 suggests the test method to determine the cracking risk of concrete and the time of cracking is presented as the main value for judging the cracking risk. Therefore, to evaluate more accurately the cracking risk of concrete and analyze the relationship between the cracking risk and the effect of aggregate, it is necessary to measure the exact net time to cracking. In this study, the net time to cracking was determined more accurately by strain measurement data and visual observation

  1. The restrain effect factor should be clearly explained.

: As you know, the shrinkage and cracking of concrete can be affected by the restraint of the aggregate. In this study, the restraint effect factor of aggregate used to numerically express the degree of influence on the restraint.

Reviewer 2 Report

The paper "Effect of Dune Sand on Drying Shrinkage Cracking of Fly Ash Concrete" touches on an important issue of effective use of dune sand in concrete. The paper is well structured and written in a clear and concise manner. There are, however, several issues with the research presented, that I would ask the authors to explain: 

  1. How was the composition of the concrete designed? Namely, what were the assumption for the composition. 
  2. Why was the coarse aggregate size chosen to be higher than 13mm, if it would need to be sieved out for some tests? This decision meant that the concrete composition is different for different tests, and thus any comparison should take it under consideration. Additionally, there is no description of coarse aggregate. 
  3. When considering the composition, fractions of dune sand smaller than 0.063mm should be treated as a part of the paste (filler), due to its fineness. This would mean that the amount of paste is subtly changing with the changes in sand content. Was this taken under considertion in the research? 
  4. There is no statistical analysis of any of the results, making it hard to judge in context of differences. Additionally, there is no clear information concering the amount of samples tested. 

Author Response

Thank you for your comments on the paper.

We considered carefully your comments and our answers are the follows.

  1. How was the composition of the concrete designed? Namely, what were the assumption for the composition. 

: In real construction projects, concrete mix design should meet the durability requirements according to the international codes such as ACI or BS. In BS 8500-1, the concrete exposed to seawater or aggressive ground should meet the W/C ratio limit to protect the corrosion of reinforcement or concrete. The lowest value of W/C ratio limit is 0.35 in BS 8500-1 but BS explain that it is not possible to produce a practical concrete with a maximum w/c ratio of 0.35 in some parts. So we selected the second lowest W/C ratio limit of 0.4. Also, the target slump was fixed at 180mm considering the slump value used in many projects in these area. The unit water content was set to two grades, 170kg/m3 and 180kg/m3. The binder content was calculated by W/C ratio and the unit weight of aggregate was determined through lab trial mix.

  1. Why was the coarse aggregate size chosen to be higher than 13mm, if it would need to be sieved out for some tests? This decision meant that the concrete composition is different for different tests, and thus any comparison should take it under consideration. Additionally, there is no description of coarse aggregate. 

: The maximum size of coarse aggregate used in this study was 20mm. This size of coarse aggregate is mainly used in these area. The size of coarse aggregate was added in manuscript. And to fabricate specimens for the ring type restrained shrinkage cracking test, the mixed concrete was sieved to remove CA that exceeded 13 mm according to test method of ASTM C 1581.

  1. When considering the composition, fractions of dune sand smaller than 0.063mm should be treated as a part of the paste (filler), due to its fineness. This would mean that the amount of paste is subtly changing with the changes in sand content. Was this taken under considertion in the research? 

: As you commented, the fines smaller than 0.063mm treated as a part of the paste (filler). However, in BS EN 12620, fine aggregate grades are divided according to the content of smaller than 0.063mm fines smaller than 0.063mm; 3%≤, 4%≤, 5%≤, 6%≤, etc. It is difficult to know the amount of fines smaller than 0.063mm in DS. However, it can be predicted that the amount of fines smaller than 0.063mm in DS is less than 3% because the particles smaller than 0.08mm in DS was 3.2%. So, in this study, DS was used only as a fine aggregate and the part of DS was not treated as a binder filler.

  1. There is no statistical analysis of any of the results, making it hard to judge in context of differences. Additionally, there is no clear information concering the amount of samples tested. 

: If the test was repeated several times, tests data such as the number of tests, average, and standard deviation could be valid. However, the concrete mixing was conducted once or twice for each mixture in a Lab. And then strength for each mixture at each age was calculated as the average value of three specimens. The data of restrained drying shrinkage test were the average value of two specimens. The number of specimen for each test was added in the manuscript. Therefore, it is hard to present the number of repetitions and the standard deviation. Please consider this situation.

Round 2

Reviewer 1 Report

The language and should  should be improved.

Author Response

Thank you very much for taking your valuable time to review my manuscript.

As you mentioned, the authors of this manuscript are not native English speakers. Therefore, we paid several times and received a premium English proofreading service from Editage (http://www.editage.co.kr). Please take this into consideration.

Reviewer 2 Report

The answers provided by the authors are interesting and explain a lot of issues that were described in the previous review round. However I would ask authors to provide information about the sample amounts in the text, if statistics are not availible, as it will be an important infromation for any reader. 

Author Response

Thanks again for commenting to improve the quality of the paper.

I understand that your comment is that the information on the number of samples should be provided. 

The answer to your comment is as follows.

For tests of compressive strength and tensile strength, three specimens were used for each mixture and two specimens were moulded for each mixture for the drying shrinkage test. 

These imformation is provided in clause 2.3 Test method in the revised manuscript; line 133 and 145

Please check it.

Best regards,